# Effect of High Hydrostatic Pressure Combined with Sous-Vide Treatment on the Quality of Largemouth Bass during Storage

**DOI:** 10.3390/foods11131931

**Published:** 2022-06-29

**Authors:** Mingzhu Zhou, Yuzhao Ling, Fangxue Chen, Chao Wang, Yu Qiao, Guangquan Xiong, Lan Wang, Wenjin Wu, Liu Shi, Anzi Ding

**Affiliations:** 1Key Laboratory of Cold Chain Logistics Technology for Agro-Product, Ministry of Agriculture and Rural Affairs, Institute of Agro-Products Processing and Nuclear Agricultural Technology, Hubei Academy of Agricultural Sciences, Wuhan 430064, China; 101900476@hbut.edu.cn (M.Z.); lingyuzhao2012@163.com (Y.L.); aphroditesnow@hotmail.com (F.C.); xiongguangquan@163.com (G.X.); lilywang_2016@163.com (L.W.); wuwenjin@hbaas.com (W.W.); shiliu@hbaas.com (L.S.); anzi@hbaas.com (A.D.); 2Key Laboratory of Fermentation Engineering (Ministry of Education), Hubei Key Laboratory of Industrial Microbiology, National “111” Center for Cellular Regulation and Molecular Pharmaceutics, Cooperative Innovation Center of Industrial Fermentation (Ministry of Education & Hubei Province), Hubei Research Center of Food Fermentation Engineering and Technology, Hubei University of Technology, Wuhan 430068, China; wangchao@hbut.edu.cn; 3School of Environmental Ecology and Biological Engineering, Wuhan Institute of Technology, Wuhan 430205, China

**Keywords:** high hydrostatic pressure, sous-vide, sensory, storage, largemouth

## Abstract

In order to estimate the effects of high hydrostatic pressure treatment at 400 MPa for 0 min and 10 min (HHP-0, HHP-10) and high hydrostatic pressure in combination with sous-vide treatment (HHP-0+SV, HHP-10+SV) on the quality of largemouth bass stored at 4 °C for 30 days, the physicochemical changes were evaluated by microbiological determinations, pH, sensory evaluation and texture analysis, and the flavour changes were analysed by solid-phase microextraction–gas chromatography–mass spectrometry (SPME-GC-MS) and amino acid automatic analyser. The results show that HHP-0+SV and HHP-10+SV treatment effectively inhibited microbiological growth and attenuated physiochemical changes (pH, sensory evaluation, flesh and texture) of largemouth bass fillets. HHP+SV treatment prolonged the storage period of largemouth bass fillets for 24 days. The content of total free amino acids in control (CK) samples was high, but HHP+SV treatment caused the loss of free amino acid content. Especially when stored for 30 days, the total free amino acid content of HHP-0+SV and HHP-10+SV was only 14.67 mg/100 g and 18.98 mg/100 g, respectively. In addition, a total of 43 volatile compounds were detected and elucidated, among which hexanal, heptaldehyde, octanal and nonanal showed a decreasing tendency in HHP groups and an increasing trend in HHP+SV groups throughout the storage.

## 1. Introduction

In many places, especially in coastal areas, fish is considered to be an excellent source of nutrition, containing more proteins than other animals [1]. Largemouth bass (Micropterus salmoides) is native to north America. Compared with other farmed fish, it is one of the most commercial fish [2]. Protein, vitamins (including vitamin A, B2, B6) and minerals (such as iron, calcium, potassium) are abundant in largemouth bass [3]. In recent years, the intake of fish has been linked with potential health benefits that can help people prevent a variety of diseases. Nowadays, there is a greater focus on safe pretreatment methods in order to keep the original flavour and nutrition of the food to the maximum extent. For example, pretreatment such as ultrasonic and HHP.

Since consumers prefer ready-to-eat products that are as natural as possible and have sufficient shelf life, it is important to consider appropriate minimal processing without altering the content and bioavailability of major nutrients [4]. In this sense, HHP, a non-thermal technology, seems to be a promising technique for preserving heat-sensitive compounds (for example, aldehydes, pyrazines and furans) and maintaining sensory quality such as colour, as well as inactivating microorganisms and enzymes [5,6]. HHP (100–800 MPa) has been shown to inactivate a wide range of microorganisms and endogenous enzymes while retaining the sensory properties and nutritional value of food [7,8]. Ye et al. [9] reported the effect of high-pressure treatment (300 MPa/20 min) on crab meat during storage at 4 °C, and indicated that the total viable counts (TVC) (5.71 log10 CFU/g) were still below the limit (6 log10 CFU/g) on day 8. Lin et al. [10] found that HHP could reduce the bacterial load and retard TVBN production, in addition, HHP conditions of ≥400 MPa for 3 min significantly extended the shelf life of hard clams during refrigerated storage. Kim et al.’s [11] results suggested that HHP treatment would be useful for inhibiting the activity of urease, thereby reducing the fishy smells from fish. However, expensive equipment and limited inactivation of spores are major restrictions to the widespread application of HHP [12].

SV temperature cooking is a food preparation process of heating samples for a long time at a lower temperature [13]. The food is heated to the central temperature of 58–95 °C in a vacuum sealing system. Compared with traditional cooking methods, this method is characterized by less efflux of soluble nutrients [14]. The sample is vacuum-packed in cling bags before heating, which reduces oxidation and prevents cross contamination. The reported advantages of using SV are the reduction of cooking losses and lipid oxidation while enhancing colour and flavour [15]. In addition, SV can improve meat tenderness [16]. Jiang et al. [17] showed that during long-term thermal processing, SV could significantly reduce the level of protein oxidation and fat oxidation, and effectively preserve the flavour and nutrition of braised meat. The cooking yield of poultry meat processed by the SV method (88.5%) was higher than that by the traditional steaming method (71.0%). The meat was found to be redder (a* values of 2.54 and 0.74, respectively) and less yellow (b* values of 15.12 and 16.49, respectively), as well as more tender [18].

Two emerging technologies, HHP and SV are regarded as effective and economical methods. The utilization of HHP or SV alone to effectively control food spoilage has been widely reported. However, reports are rare exploring the effects of HHP combined with SV on largemouth bass as far as we know. In this study, the effects of HHP and HHP+SV on the quality of largemouth bass were evaluated by measuring the indexes of microorganism, pH value, sensory, colour, texture, GC-MS and FAA.

## 2. Materials and Methods

### 2.1. Raw Materials

Cultured largemouth bass (weight of 0.38 ± 0.07 kg) were originated from Wuhan, and subsequently transported to the slaughter room alive in plastic bags filled with crushed ice within an hour. After stunning, the scales were removed. The whole bass were immediately washed with sterile water, and then cut into similar-sized fillets (about 7 cm × 5 cm × 2 cm). Fillets were randomly divided into five groups, and then fillets were subjected to HHP and SV pretreatments. Then, the fish fillets were drained, packaged separately in high-density polyethylene bags and stored at 4 °C. Three samples of each group were selected randomly for detection and analysis at day 0, 6, 12, 18, 24 and 30 of the storage.

### 2.2. SV Cooking

The vacuum-packed fish fillets were immersed in a constant temperature water bath (DF-101S, Zhengzhou Great Wall Technology Industry and Trade Co., Ltd. Zhengzhou, China), which contained 1.3 L distilled water that was pre-heated to 90 °C. The cooking process was ended until the core temperature of samples reached 90 ± 5 °C. In the light of pre-experiments, the cooking time was 15 min. Then the samples were immediately cooled to 7 ± 2 °C in an ice water bath.

### 2.3. HHP and SV Treatment

The treatment of the packaged fish fillets was implemented using high hydrostatic pressure equipment (HPPL2-600 MPa/2L, Tianjin Huatai SenMiao Biotechnology Engineering Limited by Share Ltd., TianJin, China). High hydrostatic pressure treatments were performed at 400 MPa for different times. High hydrostatic pressure treatment was to place the packaged bass slices in the high hydrostatic pressure chamber (1 L) with water as the medium (25 °C), and keep them for a certain time when the pressure reached 400 MPa (30 s). There were three replicates for each treatment, and 300 g bass slices were used in each replicate. After HHP treatment, SV treatment was carried out according to Section 2.2. Fish fillets were subjected to four kinds of treatment.

(1)High hydrostatic pressure treatment at 400 MPa for 0 min. (HHP-0)(2)High hydrostatic pressure treatment at 400 MPa for 10 min. (HHP-10)(3)High hydrostatic pressure treatment at 400 MPa for 0 min in combination with sous vide. (HHP-0+SV).(4)High hydrostatic pressure treatment at 400 MPa for 10 min in combination with sous vide. (HHP-10+SV).

### 2.4. Microbiological Determinations

A total of 10 g sample was added into 90 mL 0.85% sterilized physiological solution and the mixture was homogenized for 60 s. The microbiological evaluations were performed based on the work of Rezaeigolestani et al. [19], where serial ten-fold dilutions were made with sterilized physiological saline solution and 1 mL of the appropriately diluted sample was poured onto total viable counts (TVC) (enumerated on plate count agar (PCA) and incubated for 72 h at 30 °C).

### 2.5. pH

The pH value was recorded on a pH meter (FiveEosy, Mettler-Toledo company, Shanghai, China). Fish muscle (10 g) was thoroughly homogenized with 100 mL distilled water and soaked for 30 min, then filtered to determine the pH.

### 2.6. Sensory Evaluation

According to the description of Liu et al. [20] with some modifications, sensory analysis of flesh colour, texture, taste and aroma was conducted on a ten-point scale, completed by eight experienced panellists (four females and four males, 20 to 35 years old), in which a score of 8.0–10.0 indicated good quality, 6.0–8.0 indicated acceptable quality, 4.0–6.0 indicated unacceptable quality, and 1.0–4.0 indicated an intense dislike.

### 2.7. Texture Analysis

Texture profile analysis was measured using a Texture Analyzer (Ta. Xt 2i/50, Stability Microsystems, Surrey, UK) equipped with a specific cylindrical probe placed horizontally on a heavy platform, according to the method reported by Yu et al. [21]. The device parameters were set to trigger: force 5 g and distance 15 mm, pretest speed 5 mm/s, test speed and post-test speed were 2 mm/s and 10 mm/s. Six replicates were made from each sample under different pretreatments.

### 2.8. Headspace SPME-GC/MS Analysis

According to Shi et al. [22]. Solid-phase microextraction–gas chromatography–mass spectrometry (SPME-GC-MS) was used to detect volatile compounds. Fillets of fish (2 g) were placed into a headspace vial containing 20 mL of water and equilibrated at 40 °C for 10 min. SPME fibre (50/30 µm DVB/CAR/PDMS extractor, Supelco, Bellefonte, PA, USA) was exposed to the vial headspace for another 40 min at 40 °C. Then, the fibre was inserted into the GC and desorbed at 250 °C for 5 min. Compounds were separated on a DB-1ms capillary column (30 m × 0.25 mm × 0.25 µm, Agilent J&W, Palo Alto, CA, USA) with helium as the carrier gas at a flow rate of 1 mL min^−1^.

The temperature in the GC oven was held at 40 °C for 2 min, then increased at 2 °C/min to 90 °C, held for 5 min, then increased at 8 °C/min to a final temperature of 250 °C. The ion source was EI. The MS source and transfer line were kept at 230 °C and 280 °C, respectively. The mass spectrometry data were collected over the full scan range of 35 to 350 *m*/*z* using positive ionization with an electron energy of 70 eV. Volatile compounds can be identified by comparing the mass spectrometry results of volatile compounds with data stored in the National Institute of Standards and Technology (NIST08) spectrum database. Area normalization was used to quantify volatile compounds.

### 2.9. Free Amino Acids (FAA) Analysis

FAAs were measured according to Yu et al. [23]. Fish fillets (2 g) were homogenized with 15 mL of 5% trichloroacetic acid for 1 min. Then they were placed in a refrigerator at 4 °C for 2 h and centrifuged at 9000× *g* for 15 min with 10 mL supernatant. Taking 5 mL supernatant, pH was adjusted to 2.0 with 6 mol/L NaOH, and then volume was adjusted with distilled water to 10 mL. The extraction was filtered using a 0.22 µm filter membrane and applied to an automatic amino acid analyser (L-8900; Hitachi, Tokyo, Japan). Three parallels for each sample.

### 2.10. Statistical Analysis

All data are presented as mean ± standard deviation. The significant difference was at the level of *p* < 0.05 by DPS software analysis and Duncan’s multiple range test. All figures were obtained by Origin 2017 (OriginLab Co., Northampton, MA, USA).

## 3. Results and Discussion

### 3.1. Microbiological Analysis

The main factor limiting the shelf life of largemouth bass fillets is the development of microbial groups during the preservation. Meat products are often easily degraded by microbial activities. Figure 1 describes the changing trend in the number of TVC of largemouth bass with different treatments during storage. The results show that the TVC of largemouth bass was significantly different with different storage times and sample treatments (*p* < 0.05). The higher initial TVC for fresh largemouth bass was 4.5 log CFU/g (Figure 1). Shuai Zhuang et al. [24] found that the initial TVC of fresh largemouth bass fillets was about 4.55 log CFU/g, and the initial value was high. The TVC value of largemouth bass fillets exceeded 7 log CFU/g, which is recognized to be the maximum acceptable limit of microorganisms [25]. The TVC of fresh largemouth bass fillets on the sixth day of storage was 7.41 log CFU/g. HHP-0- and HHP-10-treated fish fillets exceeded the maximum limit (7 log CFU/g) on day 18 and 24 of storage, respectively. HHP-0+SV- and HHP-10+SV-treated fish fillets exceeded 7 log CFU/g on day 30 of storage, as shown in Figure 1. The population of the TVC increasingly elevated with storage time. Throughout the same storage time, HHP and HHP combined with SV treatment significantly reduced the number of TVC (*p* < 0.05). Ekonomou et al. [26] found that high hydrostatic pressure (HHP; 200 MPa, 15 min) could eliminate Listeria monocytogenes in trout. At the same time, the synergistic effect of HHP, liquid smoke and freezing can reduce its bacteria. Except for 20 min SV treatment at 65 °C, all temperature/time combinations used to cook turkey slices could greatly inactivate pathogenic microorganisms [27]. Compared with the control group, the shelf life of largemouth bass fillets treated with HHP-0 and HHP-10 may be extended by 12 days and 18 days, respectively, whereas the combination of HHP with SV was substantially more effective in extending the microbiological shelf life of largemouth bass fillets by 24 days.

### 3.2. pH Value Analysis

The changes in pH values for HHP, HHP+SV and the control are given in Table 1. The initial pH value of largemouth bass fillets was 6.98. Masniyom et al. [28] found that the initial pH value of fresh largemouth bass fillets was 6.80 and that the pH value was gradually increased (7.04 ± 0.01) during 2 days of storage. HHP-0, HHP-10, HHP-0+SV and HHP-10+SV showed a comparable level of pH on day 0, 6.97 ± 0.01, 6.92 ± 0.01, 6.89 ± 0.02 and 6.85 ± 0.04, respectively. However, the pH value started increasing after 6 days of storage. Presumably due to the volatile alkaline components produced by spoilage bacteria, such as alkaline amines (trimethylamine and other volatile amines) produced by fish-spoiling bacteria [29]. The pH of largemouth bass fillets was sharply decreased after one week of storage. The decrease in pH value may be due to the release of lactic acid produced by the anaerobic glycolysis of glucose and inorganic phosphate produced by ATP degradation [30]. On the 30th day of storage, HHP and HHP+SV treatment resulted in pH values of 6.4 and 6.8 for largemouth bass fillets, respectively. In this study, HHP+SV treatment contributed to keeping the pH level of largemouth bass fillets slowly changing since it may reduce the growth of bacteria that produce volatile basic components (for example, alcohols and ketones). The pH value of largemouth bass fillets treated with HHP+SV was higher than that of the control and HHP. The increase in the pH value of fish meat during cooking was owed to the formation of disulphide bonds during the cooking process.

### 3.3. Sensory Evaluation and Texture Profile Analysis

Sensory characteristics of largemouth bass are shown in Figure 2. There were significant differences between HHP and HHP+SV in the sensory scores of colour, texture, taste and aroma, while the score of the taste attribute in HHP+SV was significantly higher than in HHP. The results of sensory analysis showed that HHP+SV would have a more intense taste attribute than HHP. The important quality characteristics of bass fillets include colour, texture, juiciness, aroma and taste. Processing technology and storage have important effects on sensory attributes of bass fillets. In the present study, aroma and taste were more significant than colour, texture and juiciness in discriminating the sensory characteristics of HHP and HHP+SV. It may be that SV treatment could help produce different aroma and taste components. Sensory characteristics tended to decrease as storage time increased, but sensory scores decreased more slowly in the HHP+SV treatment compared with the HHP group. In the later stages of storage, the fish in the HHP group became darker in colour, looser in texture and worse in smell.

Texture analysis is displayed in Figure 2. The springiness, hardness, cohesiveness and chewiness of HHP+SV were significantly higher than those in HHP (*p* < 0.05). This showed a firm and elastic taste was obtained by SV pretreatment, which was popular with consumers. The initial hardness of the largemouth bass fillets from the CK group was 14,499 g. After treatment, the hardness showed a decreasing trend and gradually decreased as the storage time increased. The HHP+SV showed a lower elasticity value when compared with HHP. The springiness of bass fillets decreased with an increase in storage time and reached the minimum value at 30 days. Chewiness refers to the time required to chew food at a fixed speed to reduce it to a size suitable for swallowing [31]. Cohesiveness and chewiness exhibited the same change curves, which were complementary to hardness. This is consistent with the results of the sensory characteristics. HHP combined with SV treatment significantly reduced the hardness and chewability of fish and had a better texture, indicating that hardness and chewability are important components in this treatment.

### 3.4. Headspace SPME-GC/MS Analysis

The flavour changes of the pretreated bass fillets under storage conditions were investigated by HS-SPME GC/MS. A total of 43 volatile compounds including alcohol (10), aldehyde (10), ketone (3) hydrocarbons (17) and others (3) were detected in the sample.

As shown in Table 2, the most important compounds were alcohols and aldehydes, but alcohols were not the main contributors to the overall aroma because of their high odour threshold [32]. The odour threshold of nonanal, hexanal and other aldehydes was low. They come from the degradation of fatty acids and triglycerides, which can have a great impact on the overall aroma of marine products [33]. A few compounds such as valeraldehyde, trans-2-heptenal, cyclobutanol and 1-nonanol appeared occasionally during storage. According to the treatment and storage conditions of bass meat, most compounds can be divided into three categories. The first group included four compounds that were detected throughout the storage of bass meat. The second group included many compounds that were detected only in the middle and/or end stage, and at the end of treatment and storage; the initially detected compounds decreased or disappeared.

In the first group of compounds, hexanal, heptaldehyde, octanal and nonanal exhibited a tendency to reduce gradually in HHP groups throughout the whole storage, whereas they tended to increase in HHP+SV groups. The relative content of the second group of compounds in fish fillets was low. The species and peak areas of aldehydes were significantly reduced after HHP and UHP+SV treatment. Only the HHP-treated samples did not detect benzaldehyde, indicating that the HHP-only treatment caused the samples to lose their nutty aroma, while the HHP+SV treatment avoided this phenomenon and maintained the original flavour characteristics. After treatment and storage of the samples, trans-2-heptenal and 1-nonanol compounds disappeared. In some cases, metabolites produced during corruption may be further metabolized by microorganisms [34]. These aspects probably explain the above VOC changes. At the same time, some VOC accumulation was caused by the metabolism of spoilage microorganisms (such as *Pseudomonas*, *Shewanella* and *Enterobacteriaceae*) and chemical reactions mainly including the oxidation of polyunsaturated fatty acids [35].

### 3.5. Free Amino Acid (FAA) Analysis

Table 3 presents the effects of different treatments on the content of FAA (expressed in mg/100 g dry matter) of largemouth bass during storage. Statistical analysis indicates that HHP and HHP+SV treatment had significant effects on the total FAA content of bass (*p* < 0.05). There were significant differences among CK, HHP and HHP+SV treatments. The total amount of FAA in CK samples was high. This indicates that HHP and HHP+SV treatment would lead to the loss of free amino acid content. The total FAA content of the HHP group was higher than that of the HHP+SV group at early storage (0 day) and late storage (30 day), which indicates that SV treatment can slow down the production of amino acid molecules. In the latter stage of storage (30 day), the total FAA in the HHP+SV group decreased more than that in HHP group. In the composition of FAA, alanine was the only amino acid with the highest content in untreated samples. Higher levels of specific amino acids in HHP and HHP+SV samples compared with untreated samples may influence the perception of sweet (sweet amino acids include alanine, serine, threonine and glycine), bitter (bitter amino acids include phenylalanine, histidine, valine, methionine, isoleucine, leucine and arginine), umami (umami amino acids include glutamic and aspartic acid) and tasteless (tasteless amino acids include cysteine, lysine and tyrosine) attributes in comparison with untreated samples (Table 3). After HHP and HHP+SV treatments, the sweet and umami amino acid content of the sea bass fillets was reduced, whereas the bitter amino acid content was increased. On the 30th day of storage, the content of sweet and umami amino acids of largemouth bass decreased significantly, and the content of sweet and fresh amino acids of HHP-SV was lower than that of HHP.

## 4. Conclusions

The microbe, pH, sensory, colour, texture, GC-MS, FAA and other quality indicators of largemouth bass stored for 30 days after HHP and SV treatment were studied. The results indicated that along with the increase in storage time, the TVC of largemouth bass increased and the sensory evaluation decreased. HHP-0- and HHP-10-treated fish fillets exceeded the maximum limit (7 log CFU/g) on day 18 and 24 of storage, respectively. HHP-0+SV- and HHP-10+SV-treated fish fillets exceeded 7 log CFU/g on day 30 of storage. This showed that HHP+SV treatment can prolong the storage period of bass compared with HHP. The results of sensory evaluation, texture characteristics, GC-MS and FAA showed that HHP+SV had higher sensory and texture and good wind characteristics compared with HHP, but it would cause the loss of free amino acid content of bass. In general, HHP+SV processing is considered to be an effective technology, compared with HHP processing and SV processing alone.

## Figures and Tables

**Figure 1 foods-11-01931-f001:**
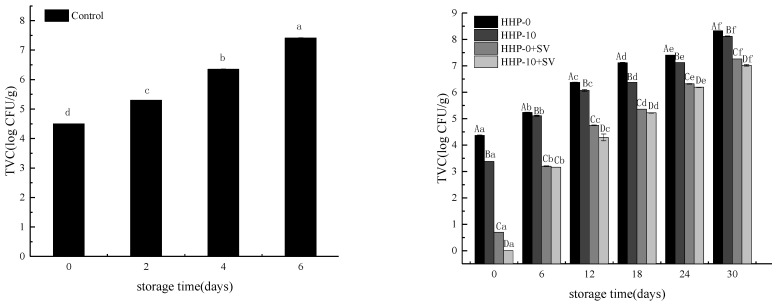
The effect of different treatments on the number of TVC of largemouth bass during storage. A–D: Differences between different treatment groups; a–f: Differences in treatment groups during storage.

**Figure 2 foods-11-01931-f002:**
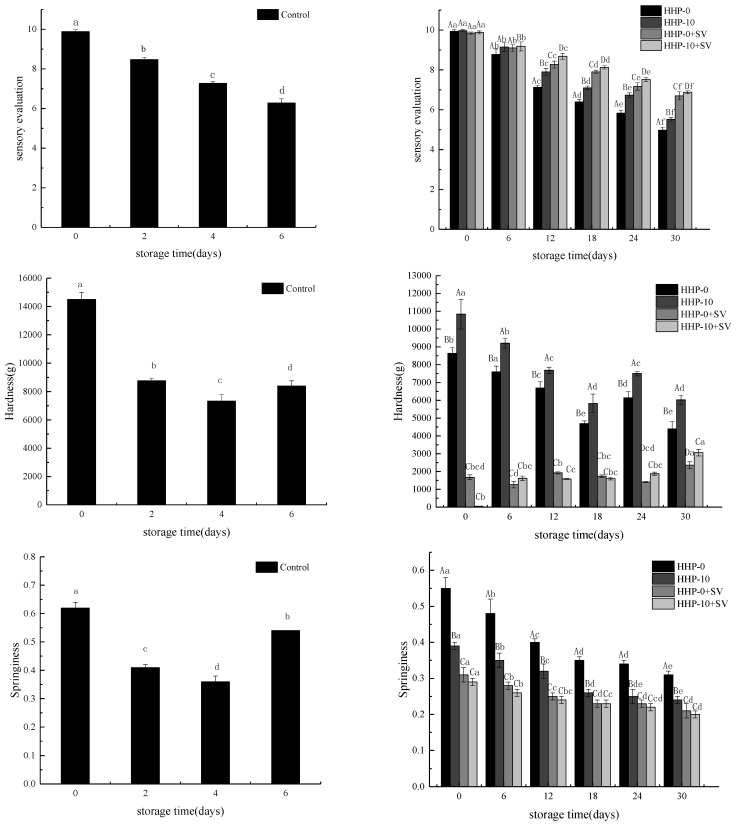
The effect of different treatments on sensory evaluations: springiness, hardness, cohesiveness and chewiness of largemouth bass during storage. A–D: Differences between different treatment groups; a–f: Differences in treatment groups during storage.

**Table 1 foods-11-01931-t001:** Effects of different treatments on the pH value of largemouth bass during storage.

Treatment	Storage Time (Days)					
	0	2	4	6	_	_
Control	6.98 ± 0.01 ^b^	7.04 ± 0.01 ^a^	6.74 ± 0.01 ^c^	6.56 ± 0.01 ^d^	_	_
**Treatment**	**Storage time (days)**					
	**0**	**6**	**12**	**18**	**24**	**30**
HHP-0	6.97 ± 0.01 ^Ab^	7.02 ± 0.01 ^Ca^	6.87 ± 0.01 ^Dc^	6.60 ± 0.01 ^Dd^	6.51 ± 0.01 ^De^	6.40 ± 0.01 ^Df^
HHP-10	6.92 ± 0.01 ^Bc^	6.97 ± 0.01 ^Da^	6.94 ± 0.01 ^Cb^	6.86 ± 0.01 ^Cd^	6.60 ± 0.01 ^Ce^	6.43 ± 0.01 ^Cf^
HHP-0+SV	6.89 ± 0.02 ^Cc^	7.12 ± 0.01 ^Aa^	7.07 ± 0.01 ^Ab^	6.89 ± 0.01 ^Bc^	6.81 ± 0.01 ^Bd^	6.75 ± 0.01 ^Be^
HHP-10+SV	6.85 ± 0.04 ^De^	7.05 ± 0.01 ^Ba^	7.01 ± 0.01 ^Bb^	6.97 ± 0.01 ^Ac^	6.89 ± 0.01 ^Ad^	6.85 ± 0.01 ^Ae^

A–D: Differences between different treatment groups; a–f: Differences in treatment groups during storage.

**Table 2 foods-11-01931-t002:** Effects of different treatments on the volatile compounds of largemouth bass during storage.

Compound (×10^6^)	0 Day	2 Day	4 Day	6 Day	0 Day				30 Day			
	CK	CK	CK	CK	HHP-0	HHP-0+SV	HHP-10	HHP-10+SV	HHP-0	HHP-0+SV	HHP-10	HHP-10+SV
Aldehydes (10)												
Benzaldehyde	0.02	0.03	-	0.02	-	0.02	-	0.03	-	0.11	-	0.04
Valeraldehyde	0.28	-	0.19	0.36	-	1.23	0.23	3.42	-	-	-	-
Isovaleraldehyde	0.17	0.63	-	0.44	-	-	0.07	0.07	0.16	-	0.28	-
Hexanal	14.73	3.13	7.68	2.38	11.40	2.57	11.34	2.27	1.50	4.04	4.12	15.02
Heptaldehyde	2.19	2.35	0.41	1.43	0.70	2.05	1.09	1.86	0.32	3.16	0.48	2.28
trans-2-Heptenal	-	0.10	0.01	0.08	0.05	-	-	-	-	-	-	-
Octanal	2.50	2.52	0.50	1.66	1.12	1.38	1.00	1.38	0.26	3.54	0.37	1.04
(E)-2-Octenal	0.07	0.21	0.02	0.22	0.04	-	-	-	-	0.07	-	-
Nonanal	3.92	5.12	1.12	3.16	2.34	1.24	1.96	0.97	1.15	3.51	0.76	0.86
Decanal	0.11	0.15	0.04	0.12	0.11	-	0.05	-	0.06	0.05	-	-
Alcohols (10)												
Cyclobutanol	-	-	-	-	0.16	-	0.17	-	-	-	-	-
1-Pentanol	0.72	1.20	0.28	0.24	0.24	0.52	0.22	0.28	0.01	1.79	0.09	-
1-Penten-3-ol	-	-	-	-	-	-	-	-	-	1.66	-	3.11
3-Methyl-1-butanol	-	-	-	-	-	-	-	0.20	11.65	0.90	3.31	-
Hexyl alcohol	1.32	1.76	0.96	1.27	1.51	-	0.54	-	0.37	5.68	-	-
n-Heptanol	-	-	-	0.91	0.53	-	0.36	-	-	-	-	-
1-Octanol	0.50	0.87	0.15	0.54	0.48	0.04	0.21	0.09	0.08	1.01	0.06	-
1-Octen-3-ol	1.89	4.39	0.67	2.90	0.90	2.09	-	1.90	0.38	9.97	1.15	9.14
1-Nonanol	-	0.15	0.01	0.03	-	-	-	-	-	-	-	-
Decyl alcohol	0.03	-	-	-	-	-	-	-	-	-	-	-
Ketones (3)												
2-Heptanone	-	0.07	-	-	-	-	-	-	0.25	0.64	-	-
2,3-Octanedione	1.69	3.58	0.43	1.75	0.72	-	0.64	0.93	0.20	0.98	0.45	0.55
2-Nonanone	0.05	0.08	-	0.04	-	0.02	-	0.04	0.03	0.14	0.37	0.10
Hydrocarbons (17)												
Cycloheptane	-	0.09	-	-	0.04	-	-	-	-	-	-	-
1-Octane	0.33	0.39	0.03	0.06	0.08	-	0.05	0.13	0.04	2.21	-	-
n-Nonane	-	-	0.01	0.02	-	-	-	0.16	0.34	-	0.85	-
Decane	-	0.09	0.00	0.08	0.01	0.05	-	0.06	0.04	0.17	0.07	0.20
Undecane	0.02	0.06	0.03	0.03	0.02	0.04	0.04	-	0.04	0.20	-	-
Dodecane	0.02	0.02	0.09	0.04	-	0.03	0.01	0.06	0.06	-	0.06	0.33
Tridecane	0.01	0.03	0.04	0.03	0.01	0.06	0.02	0.08	0.04	0.16	0.04	0.25
Tetradecane	0.03	0.04	0.03	0.02	0.02	0.08	0.02	0.13	0.04	0.09	0.11	0.29
Pentadecane	0.15	0.19	0.18	0.11	0.14	2.26	0.11	3.46	0.25	0.49	3.69	7.34
Hexadecane	0.01	0.01	0.01	0.01	-	0.02	0.01	0.02	0.01	0.02	0.03	0.06
Phenylethylene	-	-	0.17	0.04	0.04	0.07	0.03	0.05	0.02	-	0.08	0.65
Benzocyclobutene	0.01	-	0.01	0.04	-	0.17	-	0.07	-	-	0.57	1.01
1,3-Cyclooctadiene	0.19	0.07	-	-	0.18	-	-	0.02	-	-	-	3.05
1,3,5,7-Cyclooctatetraene	-	-	0.12	0.02	0.01	0.16	0.02	0.09	-	-	0.04	-
1,3,6-Octatriene,3,7-dimethyl-	0.05	0.06	-	-	-	-	-	-	-	0.21	-	-
1-Tridecene	-	-	-	-	-	0.02	-	0.06	-	-	0.03	0.10
l-Caryophyllene	-	-	-	-	-	0.02	-	0.02	-	0.10	0.03	0.07
Others (3)												
2-Pentylfuran	0.03	0.01	-	-	0.02	0.02	0.02	0.02	-	0.50	-	0.10
3-Methylpyridazine	0.02	-	-	-	-	-	-	-	-	-	-	-
Dimethyl disulfide	-	-	-	-	-	-	-	-	-	0.19	-	-

**Table 3 foods-11-01931-t003:** Effects of different treatments on the free amino acids in largemouth bass during storage.

FAA (mg/100 g)	0 Day	2 Day	4 Day	6 Day	0 Day				30 Day			
	CK				HHP-0	HHP-10	HHP-0+SV	HHP-10+SV	HHP-0	HHP-10	HHP-0+SV	HHP-10+SV
Thr	12.29 ^b^	10.10 ^e^	11.53 ^c^	9.26 ^g^	7.35 ^j^	10.38 ^d^	8.97 ^h^	14.52 ^a^	1.26 ^l^	10.02 ^f^	8.89 ^i^	7.29 ^k^
Ser	3.94 ^g^	3.83 ^h^	4.21 ^e^	4.43 ^d^	5.09 ^b^	5.33 ^a^	4.79 ^c^	3.39 ^j^	0.53 ^l^	4.10 ^f^	1.51 ^k^	3.54 ^i^
Ala	38.68 ^g^	45.27 ^b^	45.05 ^c^	40.61 ^f^	40.68 ^e^	54.27 ^a^	42.89 ^d^	11.14 ^k^	12.59 ^j^	14.99 ^i^	5.11 ^l^	21.75 ^h^
Gly	16.76 ^a^	14.57 ^c^	13.90 ^d^	13.41 ^e^	11.15 ^h^	14.57 ^c^	11.77 ^g^	13.29 ^f^	14.73 ^b^	6.69 ^j^	5.39 ^k^	7.95 ^i^
**Sweet**	71.67	73.77	74.69	67.71	64.26	84.55	68.42	42.33	29.11	35.80	20.90	40.54
Asp	0.48 ^g^	0.27 ^l^	0.32 ^j^	0.51 ^f^	0.61 ^d^	0.45 ^h^	0.77 ^b^	0.29 ^k^	0.66 ^c^	0.87 ^a^	0.55 ^e^	0.36 ^i^
Glu	5.96 ^e^	6.05 ^d^	6.88 ^b^	6.63 ^c^	4.95 ^g^	5.69 ^f^	4.28 ^i^	4.76 ^h^	11.68 ^a^	5.69 ^f^	2.37 ^k^	2.87 ^j^
**Umami**	6.44	6.32	7.20	7.14	5.56	6.13	5.05	5.05	12.34	6.56	2.92	3.23
Val	3.53 ^h^	3.70 ^e^	3.82 ^c^	3.64 ^f^	3.61 ^g^	3.95 ^b^	3.63 ^f^	4.12 ^a^	3.74 ^d^	3.40 ^i^	2.54 ^j^	2.43 ^k^
Met	2.15 ^e^	2.38 ^b^	2.43 ^a^	2.37 ^c^	1.82 ^i^	2.31 ^d^	2.03 ^g^	2.13 ^f^	2.31 ^d^	2.00 ^h^	1.35 ^j^	1.28 ^k^
Ile	2.43 ^g^	2.53 ^f^	2.74 ^d^	2.58 ^e^	2.38 ^h^	3.22 ^a^	2.98 ^c^	3.00 ^b^	1.73 ^k^	2.21 ^i^	1.51 ^l^	1.86 ^j^
Leu	4.69 ^f^	4.70 ^e^	5.16 ^d^	4.69 ^e^	4.25 ^h^	5.85 ^a^	5.38 ^c^	5.56 ^b^	3.56 ^i^	4.27 ^g^	3.13 ^k^	3.53 ^j^
Phe	3.39 ^c^	3.29 ^e^	3.42 ^b^	3.14 ^g^	3.12 ^h^	3.58 ^a^	3.33 ^d^	3.23 ^f^	3.13 ^gh^	2.23 ^i^	1.46 ^k^	1.64 ^j^
His	28.08 ^k^	27.70 ^l^	33.59 ^f^	29.72 ^j^	32.05 ^h^	42.78 ^b^	38.35 ^c^	44.14 ^a^	33.47 ^g^	36.22 ^e^	36.35 ^d^	30.46 ^i^
Arg	0.41 ^h^	0.25 ^k^	0.37 ^i^	0.62 ^a^	0.28 ^j^	0.59 ^b^	0.49 ^d^	0.58 ^c^	0.19 ^l^	0.45 ^f^	0.48 ^e^	0.43 ^g^
**Bitter**	44.68	44.55	51.53	46.76	47.51	62.29	56.19	62.76	48.14	50.78	46.82	41.64
Cys	0.00 ^f^	0.00 ^f^	0.00 ^f^	0.00 ^f^	0.00 ^f^	0.00 ^f^	0.00 ^f^	0.11 ^e^	3.53 ^a^	0.89 ^b^	0.84 ^c^	0.34 ^d^
Tyr	3.17 ^b^	3.15 ^c^	2.10 ^l^	2.72 ^h^	2.81 ^g^	3.01 ^e^	2.87 ^f^	2.70 ^i^	4.45 ^a^	3.04 ^d^	2.23 ^j^	2.20 ^k^
Lys	23.60 ^d^	21.61 ^f^	28.03 ^a^	22.89 ^e^	15.11 ^j^	21.23 ^g^	16.59 ^h^	13.72 ^k^	25.89 ^b^	24.59 ^c^	11.60 ^l^	16.44 ^i^
**Tasteless**	26.77	24.76	30.13	25.61	17.93	24.23	19.46	16.53	33.87	28.51	14.67	18.98
Total	149.56	149.40	163.55	147.22	135.26	177.21	149.11	126.67	123.47	121.66	85.31	104.38

Note: letters represent significant differences. *p* < 0.05.

## Data Availability

Data is contained within the article.

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
