# Peer review of "Effect of High Hydrostatic Pressure Combined with Sous-Vide Treatment on the Quality of Largemouth Bass during Storage"

_foods, 2022, doi:10.3390/foods11131931_

Round 1

Reviewer 1 Report

This manuscript describes the effect of high-pressure combined with Sous-vide treatment on the quality of largemouth bass during storage. Only two replicates were used, so the scope is somewhat limited, and the authors should weaken all the statements which attempt a generalisation. While a relatively large number of volatiles was identified using SPME-GC-MS, given the limited number of media (only PCA for TVC) used for isolation and the poor discussion of the results, there are no clear conclusions. Said that a sample should be taken at least before “spoilage” and for “spoiled or spoiling samples” to be sufficient to detect the VOCs associated with the treatments. For the manuscript to become acceptable for publication, the authors should present all the materials and methods in detail and discuss their results and the advantages and shortcomings of their approach. I am under the impression that the aim of the current study is not completely clear. Finally, the authors should write a clear conclusion to present the importance of their findings.

Reviewer 2 Report

Effect of ultrahigh-pressure combined with sous-vide treatment on the quality of largemouth bass during storage

This manuscript is well organized and I do not see any large mistakes or problems. I have only minor comments.

  • Line 81 I cannot find on web Hyperbaric equipment. I know Hiperbaric company
  • Line 109 There is need to generate the sentence like “Measuring was made etc.”.
  • Line 141 there is also need to generate the sentence.
  • Figure1 has to be placed in text to enable text to be seen not to cover is by figure.
  • There are no codes In tables 2 and 3 to differ results as a function of storage time by statistical evaluation.
  • References can be enriched by some classical publications dealing with UHP, not only papers but existing books.

Reviewer 3 Report

Effect of Ultrahigh-pressure combined with Sous-vide treatment on the quality of largemouth bass during storage

Comments:

Line no 26: CK? Use full name then abbreviate it.

Line no 31: Abstract is well written and explanatory.

Line no 35: Mention the types of protein.

Line no 37: Which protein make fish profile superior than other animals?

Line no 41: Comparison is on the basis of?

Line no 42: Pretreatment method like? Give example

Line no 44-46. Please check these paper on high pressure in link with high pressure as clean label technologies, it would increase the importance of your paper if you read these article and a few line here.  

  1. LWT-Food Science and Technology, 141(9), 111828.
  2. Food Research International, 150, 110792.
  3. Innovations in high-pressure technologies for the development of clean label dairy products: A review. Food Reviews International.

Line no 47: Mention heat sensitive compounds.

Line no 48. Please check this paper for enzyme reduction through high pressure (Impact of High-Pressure treatments on Enzyme activity of Fruit Based Beverages: An Overview. International Journal of Food Science and Technology, 57(2), 801-815).

Line no 51. TVC means?

Line no 53. What is CV?

Line no 56: SV mechanisms need to discuss briefly.

Line no 133: Which microbial group affects the meat commonly?

Line no 156: Graphical work is appreciable.

Line no 175: Does decrease in pH affects the quality positively?

Line no 179: Which volatile components liberate?

Line no 202: Which components made the combination treatment more effective in context of texture profile? Discuss

Line no 247: what does it mean sweet and bitter amino acids? Unable to get idea

  • Section no 3.5 is too short it should be more comprehensive so your research results can be interpreted.
  • Conclusion is too short and not justifies your manuscript kindly do some work on these points.
  • All the abbreviations should be properly written for complete at first time and then at abbreviated form.

Round 2

Reviewer 1 Report

The manuscript has been improved significantly since the last time. However,  I would suggest the authors to clearly mention in the manuscript what they mean by the 0-time high-pressure treatment (necessary) and include in the materials and methods the keep time for the pressure to reach 400 MPa.

Finally, the authors should mention why high-pressure treatment and sous vide are considered "emerging" technologies if they need to use this term in the introduction section and throughout the text.

Author Response

Thank you very much for your valuable comments. I will benefit a lot.

Point 1: The manuscript has been improved significantly since the last time. However,  I would suggest the authors to clearly mention in the manuscript what they mean by the 0-time high-pressure treatment (necessary) and include in the materials and methods the keep time for the pressure to reach 400 MPa.

Response 1: Thanks for your suggestion. In lines 91-95 of the article, the holding time of ultra-high pressure (400MPa) has been explained, and 0min has also been mentioned.

Point 2: Finally, the authors should mention why high-pressure treatment and sous vide are considered "emerging" technologies if they need to use this term in the introduction section and throughout the text.

Response 2: Thanks for your suggestion. In the introduction part of the article, UHP seems to be a promising technique for preserving heat-sensitive compounds (For example. Aldehydes, pyrazines and furans) and maintaining sensory quality such as colour , as well as inactivating microorganisms and enzyme mechanisms. The reported advantage of using SV is the reduction of cooking losses and lipid oxidation while enhancing color and flavor. In addition, SV can improve meat tenderness. Therefore, UHP and SV are regarded as emerging technologies.

Reviewer 3 Report

Well revised. 

Author Response

Thank you very much for your valuable comments, which will give me better guidance in writing my report in the future.